# VQ-WAV2VEC: SELF-SUPERVISED LEARNING OF DISCRETE SPEECH REPRESENTATIONS

**Alexei Baevski**[*△]    **Steffen Schneider**[*▽†]    **Michael Auli**[△]

[△] Facebook AI Research, Menlo Park, CA, USA
[▽] University of Tübingen, Germany

## ABSTRACT

We propose vq-wav2vec to learn discrete representations of audio segments through a wav2vec-style self-supervised context prediction task. The algorithm uses either a Gumbel-Softmax or online k-means clustering to quantize the dense representations. Discretization enables the direct application of algorithms from the NLP community which require discrete inputs. Experiments show that BERT pre-training achieves a new state of the art on TIMIT phoneme classification and WSJ speech recognition.[1]

## 1 INTRODUCTION

Learning discrete representations of speech has gathered much recent interest (Versteegh et al., 2016; Dunbar et al., 2019). A popular approach to discover discrete units is via autoencoding (Tjandra et al., 2019; Eloff et al., 2019; Chorowski et al., 2019) sometimes coupled with an autoregressive model (Chung et al., 2019). Another line of research is to learn continuous speech representations in a self-supervised way via predicting context information (Chung & Glass, 2018; van den Oord et al., 2018; Schneider et al., 2019).

In this paper, we combine these two lines of research by learning discrete representations of speech via a context prediction task instead of reconstructing the input. This enables us to directly apply well performing NLP algorithms to speech data (Figure 1a).

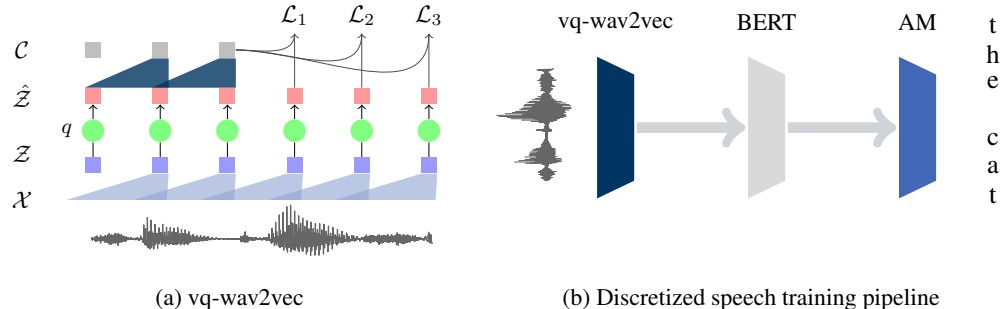

(a) vq-wav2vec                    (b) Discretized speech training pipeline

Figure 1: (a) The vq-wav2vec encoder maps raw audio ($\mathcal{X}$) to a dense representation ($\mathcal{Z}$) which is quantized (q) to $\hat{\mathcal{Z}}$ and aggregated into context representations ($\mathcal{C}$); training requires future time step prediction. (b) Acoustic models are trained by quantizing the raw audio with vq-wav2vec, then applying BERT to the discretized sequence and feeding the resulting representations into the acoustic model to output transcriptions.

Our new discretization algorithm, vq-wav2vec, learns discrete representations of fixed length segments of audio signal by utilizing the wav2vec loss and architecture (Schneider et al, 2019; §2). To

---

[*]Equal contribution.
[†]Work done during a Facebook AI residency.
[1]The code will be made available at `http://github.com/pytorch/fairseq`.

choose the discrete variables, we consider a Gumbel-Softmax approach (Jang et al., 2016) as well as online k-means clustering, similar to VQ-VAE (Oord et al., 2017; Eloff et al., 2019; §3).

We then train a Deep Bidirectional Transformer (BERT; Devlin et al., 2018; Liu et al., 2019) on the discretized unlabeled speech data and input these representations to a standard acoustic model (Figure 1b; §4). Our experiments show that BERT representations perform better than log-mel filterbank inputs as well as dense wav2vec representations on both TIMIT and WSJ benchmarks. Discretization of audio enables the direct application of a whole host of algorithms from the NLP literature to speech data. For example, we show that a standard sequence to sequence model from the NLP literature can be used to perform speech recognition over discrete audio tokens (§5, §6).

## 2 BACKGROUND

### 2.1 WAV2VEC

wav2vec (Schneider et al., 2019) learns representations of audio data by solving a self-supervised context-prediction task with the same loss function as word2vec (Mikolov et al., 2013; van den Oord et al., 2018). The model is based on two convolutional neural networks where the the *encoder* produces a representation $\mathbf{z}_i$ for each time step $i$ at a rate of 100 Hz and the *aggregator* combines multiple encoder time steps into a new representation $\mathbf{c}_i$ for each time step $i$. Given an aggregated representation $\mathbf{c}_i$, the model is trained to distinguish a sample $\mathbf{z}_{i+k}$ that is $k$ steps in the future from distractor samples $\tilde{\mathbf{z}}$ drawn from a distribution $p_n$, by minimizing the contrastive loss for steps $k = 1, \ldots, K$:

$$\mathcal{L}_k^{\text{wav2vec}} = -\sum_{i=1}^{T-k} \left( \log \sigma(\mathbf{z}_{i+k}^\top h_k(\mathbf{c}_i)) + \lambda \mathop{\mathbb{E}}_{\tilde{\mathbf{z}} \sim p_n} [\log \sigma(-\tilde{\mathbf{z}}^\top h_k(\mathbf{c}_i))] \right) \tag{1}$$

where $T$ is the sequence length, $\sigma(x) = 1/(1 + \exp(-x))$, and where $\sigma(\mathbf{z}_{i+k}^\top h_k(\mathbf{c}_i))$ is the probability of $\mathbf{z}_{i+k}$ being the true sample. We consider a step-specific affine transformation $h_k(\mathbf{c}_i) = W_k \mathbf{c}_i + \mathbf{b}_k$ that is applied to $\mathbf{c}_i$ (van den Oord et al., 2018). We optimize the loss $\mathcal{L} = \sum_{k=1}^{K} \mathcal{L}_k$, summing (1) over different step sizes. After training, the representations produced by the context network $\mathbf{c}_i$ are input to the acoustic model instead of log-mel filterbank features.

### 2.2 BERT

BERT (Devlin et al., 2018) is a pre-training approach for NLP tasks, which uses a transformer encoder model to build a representation of text. Transformers uses self-attention to encode the input sequence as well as an optional source sequence (Vaswani et al., 2017). The original BERT model combined two tasks for training: first, masked language modeling randomly removes some of the input tokens and the model has to predict those missing tokens. Second, next sentence prediction splices two different text passages together into a single example and the model needs to predict whether the passages are from the same document.

## 3 VQ-WAV2VEC

Our approach, vq-wav2vec, learns vector quantized (VQ) representations of audio data using a future time-step prediction task. We follow the same architectual choices as wav2vec (§2.1) with two convolutional networks $f : \mathcal{X} \mapsto \mathcal{Z}$ and $g : \hat{\mathcal{Z}} \mapsto \mathcal{C}$ for feature extraction and aggregation, as well as a new *quantization* module $q : \mathcal{Z} \mapsto \hat{\mathcal{Z}}$ to build discrete representations (Figure 1a).

We first map 30ms segments of raw speech to a dense feature representation $\mathbf{z}$ at a stride of 10ms using the encoder network $f$. Next, the quantizer ($q$) turns these dense representations into discrete indices which are mapped to a reconstruction $\hat{\mathbf{z}}$ of the original representation $\mathbf{z}$. We feed $\hat{\mathbf{z}}$ into the aggregator $g$ and optimize the same context prediction task as wav2vec outlined in §2.1.

The quantization module replaces the original representation $\mathbf{z}$ by $\hat{\mathbf{z}} = \mathbf{e}_i$ from a fixed size codebook $\mathbf{e} \in \mathbb{R}^{V \times d}$ which contains $V$ representations of size $d$. We consider the Gumbel-Softmax which is a differentiable approximation of the argmax for computing one-hot representations (§3.1; Figure 2a)

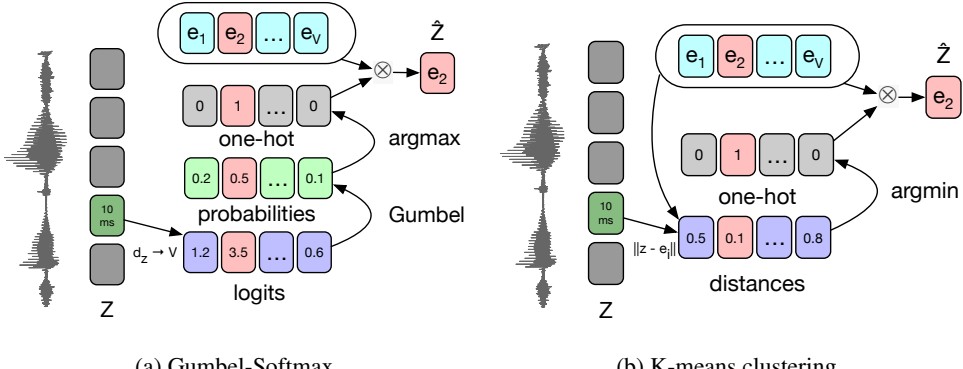

(a) Gumbel-Softmax          (b) K-means clustering.

Figure 2: (a) The Gumbel-Softmax quantization computes logits representing the codebook vectors (**e**). In the forward pass the argmax codeword (**e₂**) is chosen and for backward (not shown) the exact probabilities are used. (b) K-means vector quantization computes the distance to all codeword vector and chooses the closest (argmin).

as well as online k-means clustering, similar to the vector quantized variational autoencoder (VQ-VAE; Oord et al., 2017; §3.2; Figure 2b). Finally, we perform multiple vector quantizations over different parts of **z** to mitigate mode collapse (§3.3).

## 3.1 GUMBEL-SOFTMAX

The Gumbel-Softmax (Gumbel, 1954; Jang et al., 2016; Maddison et al., 2014) enables selecting discrete codebook variables in a fully differentiable way and we use the straight-through estimator of Jang et al. (2016). Given the dense representation **z**, we apply a linear layer, followed by a ReLU and another linear which outputs $\mathbf{l} \in \mathbb{R}^V$ logits for the Gumbel-Softmax. At inference, we simply pick the largest index in $l$. At training, the output probabilities for choosing the $j$-th variable are

$$p_j = \frac{\exp(l_j + v_j)/\tau}{\sum_{k=1}^{V} \exp(l_k + v_k)/\tau}, \tag{2}$$

where $v = -\log(-\log(u))$ and $u$ are uniform samples from $\mathcal{U}(0,1)$. During the forward pass, $i = \mathrm{argmax}_j p_j$ and in the backward pass, the true gradient of the Gumbel-Softmax outputs is used.

## 3.2 K-MEANS

The vector quantization approach of van den Oord et al. (2017) is an alternative to making the index selection procedure fully differentiable. Different to their setup, we optimize a future time step prediction loss instead of the reconstruction loss of an autoencoder.

We choose the codebook variable representation by finding the closest variable to the input features **z** in terms of the Euclidean distance, yielding $i = \mathrm{argmin}_j \|\mathbf{z} - \mathbf{e}_j\|_2^2$. During the forward pass, we select $\hat{\mathbf{z}} = \mathbf{e}_i$ by choosing the corresponding variable from the codebook. We obtain gradients for the encoder network by back-propagating $d\mathcal{L}^{\mathrm{wav2vec}}/d\hat{\mathbf{z}}$ (van den Oord et al., 2017). The final loss has two additional terms:

$$\mathcal{L} = \sum_{k=1}^{K} \mathcal{L}_k^{\mathrm{wav2vec}} + \left( \|\mathrm{sg}(\mathbf{z}) - \hat{\mathbf{z}}\|^2 + \gamma \|\mathbf{z} - \mathrm{sg}(\hat{\mathbf{z}})\|^2 \right), \tag{3}$$

where $\mathrm{sg}(x) \equiv x, \frac{\mathrm{d}}{\mathrm{d}x}\mathrm{sg}(x) \equiv 0$ is the stop gradient operator and $\gamma$ is a hyperparameter. The first term is the future prediction task and gradients do not change the codebook because of the straight-through gradient estimation of mapping **z** to $\hat{\mathbf{z}}$. The second term $\|\mathrm{sg}(\mathbf{z}) - \hat{\mathbf{z}}\|^2$ moves the codebook vectors closer to the encoder output, and the third term $\|\mathbf{z} - \mathrm{sg}(\hat{\mathbf{z}})\|^2$ makes sure that the encoder outputs are close to a centroid (codeword).

### 3.3 VECTOR QUANTIZATION WITH MULTIPLE VARIABLE GROUPS

So far, we considered replacing the encoder feature vector $\mathbf{z}$ by a single entry $\mathbf{e}_i$ in the codebook. This is prone to mode collapse where only some of the codewords are actually used. Previously, this problem has been mitigated by workarounds such as re-initializing codewords or applying additional regularizers to the loss function (Caron et al., 2019). In the following, we describe another strategy where we independently quantize partitions of $\mathbf{z}$, similar to product quantization (Jegou et al., 2011). This results in larger dictionaries and increased downstream performance (Appendix A).

The dense feature vector $\mathbf{z} \in \mathbb{R}^d$ is first organized into multiple *groups* $G$ into the matrix form $\mathbf{z}' \in \mathbb{R}^{G \times (d/G)}$. We then represent each row by an integer index, and hence can represent the full feature vector by the indices $\mathbf{i} \in [V]^G$, where $V$ again denotes the possible number of *variables* for this particular group and each element $\mathbf{i}_j$ corresponds to a fixed codebook vector. For each of the $G$ groups, we apply either one of the two VQ approaches (§3.1 and §3.2).

The codebook itself can be initialized in two possible ways: Codebook variables can be shared across groups, i.e., a particular index in group $j$ would reference the same vector as the same index in group $j'$. This yields a codebook $\mathbf{e} \in \mathbb{R}^{V \times (d/G)}$. In contrast, not sharing the codebook variables yields a codebook of size $\mathbf{e} \in \mathbb{R}^{V \times G \times (d/G)}$. In practise, we observe that sharing the codebook variables generally yields competitive results to a non-shared representation.

## 4 BERT PRE-TRAINING ON QUANTIZED SPEECH

Once we trained a vq-wav2vec model we can discretize audio data and make it applicable to algorithms that require discrete inputs. One possibility is to use the discretized training data and apply BERT pre-training where the task is to predict masked input tokens based on an encoding of the surrounding context (Devlin et al., 2018). Once the BERT model is trained, we can use it to build representations and feed them into an acoustic model to improve speech recognition. We follow recent advances in BERT training which only use the masked input token prediction (Liu et al., 2019).

Since each of the discretized tokens represents around 10 ms of audio it is likely too easy to predict a single masked input token. We therefore change BERT training by masking *spans* of consecutive discretized speech tokens, similar to Joshi et al. (2019). To mask the input sequence, we randomly sample $p = 0.05$ of all tokens to be a starting index, without replacement, and mask $M = 10$ consecutive tokens from every sampled index; spans may overlap. This makes the masked token prediction harder and we show later that it improves accuracy over masking individual tokens (§6.5).

## 5 EXPERIMENTAL SETUP

### 5.1 DATASETS

We generally pre-train vq-wav2vec and BERT on the full 960h of Librispeech (Panayotov et al., 2015) and after vq-wav2vec training it is discretized to 345M tokens. Where indicated we perform ablations on a clean 100h subset which is discretized to 39.9M tokens. We evaluate models on two benchmarks: TIMIT (Garofolo et al., 1993b) is a 5h dataset with phoneme labels and Wall Street Journal (WSJ; Garofolo et al. 1993a) is a 81h dataset for speech recognition. For TIMIT, we apply the standard evaluation protocol and consider 39 different phonemes. For WSJ, we train acoustic models directly on 31 graphemes, including the English alphabet, the apostrophe, the silence token and tokens for repeating characters.

### 5.2 VQ-WAV2VEC

We adapt the fairseq implementation of wav2vec (Schneider et al., 2019; Ott et al., 2019) and use vq-wav2vec/wav2vec models with $34 \times 10^6$ parameters. The encoder has 8 layers with 512 channels each, kernel sizes (10,8,4,4,4,1,1,1) and strides (5,4,2,2,2,1,1,1), yielding a total stride of 160. Each layer contains a convolution, followed by dropout, group normalization with a single group (Wu & He, 2018) and a ReLU non-linearity. The aggregator is composed of 12 layers, with 512 channels, stride 1, and kernel sizes starting at 2 and increasing by 1 for every subsequent layer. The block

structure is the same as for the encoder network, except we introduce skip connections between each subsequent block.

We train with the wav2vec context prediction loss (Equation 1) for 400k updates, predicting $K = 8$ steps into the future and sample 10 negatives from the same audio example. Training is warmed up for 500 steps where the learning rate is increased from $1 \times 10^{-7}$ to $5 \times 10^{-3}$, and then annealed to $1 \times 10^{-6}$ using a cosine schedule (Loshchilov & Hutter, 2016). The batch size is 10, and we crop a random section of 150k frames for each example (approximately 9.3 seconds for 16kHz sampling rate). All models are trained on 8 GPUs.

For ablations and experiments on the 100h Librispeech subset, we use a smaller model with kernels (10,8,4,4,4) and strides (5,4,2,2,2) in the encoder and seven convolutional layers with stride one and kernel size three in the aggregator. This model is trained for 40k updates.

**Gumbel-Softmax Models.** We use $G = 2$ groups and $V = 320$ latents per group and the linear layer projects the features produced by the encoder into $G \cdot V = 640$ logits. The Gumbel-Softmax produces a one-hot vector for each group $G$. The temperature $\tau$ is linearly annealed from 2 to 0.5 over the first 70% of updates and then kept constant at 0.5. This enables the model to learn which latents work best for each input before committing to a single latent. After training this model on 960h of Librispeech and quantizing the training dataset, we are left with 13.5k unique codewords combinations (out of $V^G$ = 102k possible codewords).

**k-means Models.** We use $G = 2$ groups and $V = 320$ variables per group. vq-wav2vec on full Librispeech yields 23k unique codewords. Following van den Oord et al. (2017), we found $\gamma = 0.25$ to be a robust choice for balancing the VQ auxiliary loss.

### 5.3 BERT

**BERT base** models have 12 layers, model dimension 768, inner dimension (FFN) 3072 and 12 attention heads (Devlin et al., 2018). The learning rate is warmed up over the first 10,000 updates to a peak value of $1 \times 10^{-5}$, and then linearly decayed over a total of 250k updates. We train on 128 GPUs with a batch size of 3072 tokens per GPU giving a total batch size of 393k tokens (Ott et al., 2018). Each token represents 10ms of audio data.

**BERT small**. For ablations we use a smaller setup with model dimension 512, FFN size 2048, 8 attention heads and dropout 0.05. Models are trained for 250k updates with a batch size of 2 examples per GPU.

### 5.4 ACOUSTIC MODEL

We use wav2letter as accoustic model (Collobert et al., 2016; 2019) and train for 1,000 epochs on 8 GPUs for both TIMIT and WSJ using the auto segmentation criterion. For decoding the emissions from the acoustic model on WSJ we use a lexicon as well as a separate language model trained on the WSJ language modeling data only. We consider a 4-gram KenLM language model (Heafield et al., 2013) and a character based convolutional language model (Likhomanenko et al., 2019) and tune the models with the same protocol as Schneider et al. (2019).

## 6 RESULTS

### 6.1 WSJ SPEECH RECOGNITION

We first evaluate on the WSJ speech recognition benchmark. We train a vq-wav2vec model on the unlabeled version of Librispeech, then discretize the same data with the resulting model to estimate a BERT model. Finally, we train a wav2letter acoustic model on WSJ by inputting either the BERT or vq-wav2vec representations instead of log-mel filterbanks.[2]

We compare to various results from the literature, including wav2vec (Schneider et al., 2019) and we consider three setups: performance without any language model (No LM), with an n-gram LM

---

[2]For vq-wav2vec we input the dense representations corresponding to the learned discrete units.

| | nov93dev | | nov92 | |
| --- | --- | --- | --- | --- |
| | LER | WER | LER | WER |
| Deep Speech 2 (12K h labeled speech; Amodei et al., 2016) | - | 4.42 | - | 3.1 |
| Trainable frontend (Zeghidour et al., 2018) | - | 6.8 | - | 3.5 |
| Lattice-free MMI (Hadian et al., 2018) | - | 5.66[†] | - | 2.8[†] |
| Supervised transfer-learning (Ghahremani et al., 2017) | - | 4.99[†] | - | 2.53[†] |
| NO LM | | | | |
| Baseline (log-mel) | 6.28 | 19.46 | 4.14 | 13.93 |
| wav2vec (Schneider et al., 2019) | 5.07 | 16.24 | 3.26 | 11.20 |
| vq-wav2vec Gumbel | 7.04 | 20.44 | 4.51 | 14.67 |
| + BERT base | **4.13** | **13.40** | **2.62** | **9.39** |
| 4-GRAM LM (Heafield et al., 2013) | | | | |
| Baseline (log-mel) | 3.32 | 8.57 | 2.19 | 5.64 |
| wav2vec (Schneider et al., 2019) | 2.73 | 6.96 | 1.57 | 4.32 |
| vq-wav2vec Gumbel | 3.93 | 9.55 | 2.40 | 6.10 |
| + BERT base | **2.41** | **6.28** | **1.26** | **3.62** |
| CHAR CONVLM (Likhomanenko et al., 2019) | | | | |
| Baseline (log-mel) | 2.77 | 6.67 | 1.53 | 3.46 |
| wav2vec (Schneider et al., 2019) | 2.11 | 5.10 | 0.99 | 2.43 |
| vq-wav2vec Gumbel + BERT base | **1.79** | **4.46** | **0.93** | **2.34** |

Table 1: WSJ accuracy of vq-wav2vec on the development (nov93dev) and test set (nov92) in terms of letter error rate (LER) and word error rate (WER) without language modeling (No LM), a 4-gram LM and a character convolutional LM. vq-wav2vec with BERT pre-training improves over the best wav2vec model (Schneider et al., 2019).

| | nov93dev | | nov92 | |
| --- | --- | --- | --- | --- |
| | LER | WER | LER | WER |
| NO LM | | | | |
| wav2vec (Schneider et al., 2019) | 5.07 | 16.24 | 3.26 | 11.20 |
| vq-wav2vec Gumbel | 7.04 | 20.44 | 4.51 | 14.67 |
| + BERT small | 4.52 | 14.14 | 2.81 | 9.69 |
| vq-wav2vec k-means (39M codewords) | 5.41 | 17.11 | 3.63 | 12.17 |
| vq-wav2vec k-means | 7.33 | 21.64 | 4.72 | 15.17 |
| + BERT small | 4.31 | 13.87 | 2.70 | 9.62 |
| 4-GRAM LM (Heafield et al., 2013) | | | | |
| wav2vec (Schneider et al., 2019) | 2.73 | 6.96 | 1.57 | 4.32 |
| vq-wav2vec Gumbel | 3.93 | 9.55 | 2.40 | 6.10 |
| + BERT small | 2.67 | 6.67 | 1.46 | 4.09 |
| vq-wav2vec k-means (39M codewords) | 3.05 | 7.74 | 1.71 | 4.82 |
| vq-wav2vec k-means | 4.37 | 10.26 | 2.28 | 5.71 |
| + BERT small | 2.60 | 6.62 | 1.45 | 4.08 |

Table 2: Comparison of Gumbel-Softmax and k-means vector quantization on WSJ (cf. Table 1).

(4-gram LM) and with a character convolutional LM (Char ConvLM). We report the accuracy of wav2letter with log-mel filterbanks as input (Baseline) and wav2vec. For vq-wav2vec we first experiment with the Gumbel-Softmax, with and without a BERT base model (§5.3).

Table 1 shows that vq-wav2vec together with BERT training can achieve a new state of the art of 2.34 WER on nov92. Gains are largest when no language model is used which is the fastest setting. vq-wav2vec with Gumbel-Softmax uses only 13.5k distinct codewords to represent the audio signal and this limited set of codewords is not sufficient to outperform the baseline. However, it does enable training BERT models which require a relatively small vocabulary.

|  | dev PER | test PER |
|---|---|---|
| CNN + TD-filterbanks (Zeghidour et al., 2018) | 15.6 | 18.0 |
| Li-GRU + fMLLR (Ravanelli et al., 2018) | – | 14.9 |
| wav2vec (Schneider et al., 2019) | 12.9 | 14.7 |
| Baseline (log-mel) | 16.9 | 17.6 |
| vq-wav2vec, Gumbel | 15.34 | 17.78 |
| + BERT small | 9.64 | **11.64** |
| vq-wav2vec, k-means | 15.65 | 18.73 |
| + BERT small | 9.80 | 11.40 |

Table 3: TIMIT phoneme recognition in terms of phoneme error rate (PER). All our models use the CNN-8L-PReLU-do0.7 architecture (Zeghidour et al., 2018).

|  | dev clean | dev other | test clean | test other |
|---|---|---|---|---|
| Mohamed et al. (2019) | 4.8 | 12.7 | 4.7 | 12.9 |
| Irie et al. (2019) | 4.4 | 13.2 | 4.7 | 13.4 |
| Park et al. (2019) | 2.8 | 6.8 | 2.5 | 5.8 |
| vq-wav2vec Gumbel + Transformer Big | 5.6 | 15.5 | 6.2 | 18.2 |

Table 4: Librispeech results for a standard sequence to sequence model trained on discretized audio without BERT pre-training and results from the literature. All results are without a language model.

Next, we compare Gumbel-Softmax to k-means for vector quantization. For this experiment we use the faster to train BERT small configuration (§5.3). We also train a vq-wav2vec k-means model with a very large number of codewords (39.9M) to test whether a more expressive model can close the gap to wav2vec. Table 2 shows that Gumbel-Softmax and k-means clustering perform relatively comparably: in the no language model setup without BERT, Gumbel-Softmax is more accurate than k-means but these differences disappear with BERT. For 4-gram LM setup, k-means is better but those differences disappear again after BERT training. Finally, the large codeword model can substantially reduce the gap to the original wav2vec model.

## 6.2 TIMIT Phoneme Recognition

Next, we experiment on the much smaller TIMIT phoneme recognition task where we also pre-train vq-wav2vec on the full Librispeech corpus. Table 3 shows that vq-wav2vec and BERT achieve a new state of the art of 11.64 PER which corresponds to a 21% reduction in error over the previous best result of wav2vec.

## 6.3 Sequence to Sequence Modeling

So far we used vq-wav2vec to train BERT on discretized speech. However, once the audio is discretized we can also train a standard sequence to sequence model to perform speech recognition. In preliminary experiments, we trained an off-the-shelf Big Transformer (Vaswani et al., 2017; Ott et al., 2019) on the vq-wav2vec Gumbel-Softmax discretized Librispeech corpus and evaluated on the Librispeech dev/test sets; we use a 4k BPE output vocabulary (Sennrich et al., 2016). Table 4 shows that results are promising, even though they are not as good as the state of the art (Park et al., 2019) which depends on data augmentation that we do not use.

## 6.4 Accuracy vs. Bitrate

Next, we investigate how well vq-wav2vec can compress the audio data. Specifically, we train models with different numbers of groups $G$ and variables $V$ to vary the size of the possible codebook size $V^G$ and measure accuracy on TIMIT phoneme recognition without BERT training.

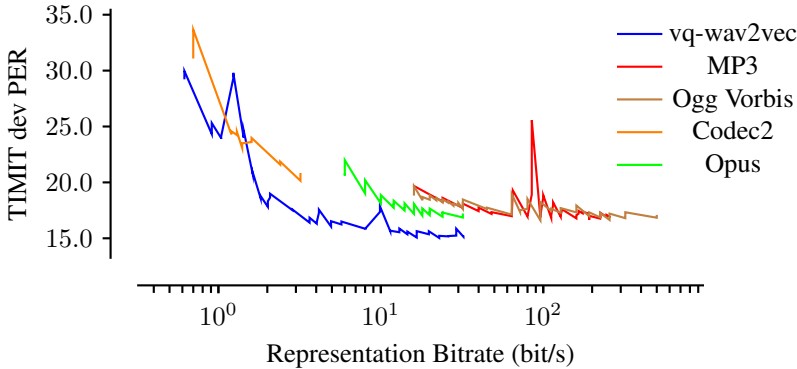

Figure 3: Comparison of PER on the TIMIT dev set for various audio codecs and vq-wav2vec k-means trained on Librispeech 100h.

We measure compression with the bitrate $r \cdot G \log_2 V$ at sampling rate $r = 100$Hz and report the trade-off between bitrate and accuracy on our phoneme recognition task. We experiment with vq-wav2vec k-means and train models with 1,2,4,8,16 and 32 groups, using 40,80,160,...,1280 variables, spanning a bitrate range from 0.53 kbit/s (G = 1, V = 40) to 33.03 kbit/s (G = 32, V = 1280). We place the quantization module after the aggregator module and train all models in the small vq-wav2vec setup (§5.2) on the 100h clean Librispeech subset.

As baselines, we consider various lossy compression algorithms applied to the TIMIT audio data and train wav2letter models on the resulting audio: Codec2[3] as a low bitrate codec, Opus (Terriberry & Vos, 2012) as a medium bitrate codec and MP3 and Ogg Vorbis (Montgomery, 2004) as high bitrate codecs. We use the whole spectrum of both variable and constant bitrate settings of the codecs; we encode and decode with ffmpeg (ffmpeg developers, 2016). Figure 3 shows the trade-off between the bitrate and TIMIT accuracy. Acoustic models on vq-wav2vec achieve the best results across most bitrate settings.

## 6.5 ABLATIONS

Table 5a shows that masking entire spans of tokens performs significantly better than individual tokens ($M = 1$). Furthermore, BERT training on discretized audio data is fairly robust to masking large parts of the input (Table 5b).

| $M$ | dev | test |
|---|---|---|
| 1 | 14.94 | 17.38 |
| 5 | 13.62 | 15.78 |
| 10 | 12.65 | 15.28 |
| 20 | 13.04 | 15.56 |
| 30 | 13.18 | 15.64 |

(a) Mask length.

| $p$ | dev | test |
|---|---|---|
| 0.015 | 12.65 | 15.28 |
| 0.020 | 12.51 | 14.43 |
| 0.025 | 12.16 | 13.96 |
| 0.030 | 11.68 | 14.48 |
| 0.050 | 11.45 | 13.62 |

(b) Mask probabilities.

Table 5: TIMIT PER for (a) different mask sizes $M$ with $pM = 0.15$ in BERT training and (b) mask probabilities $p$ for a fixed mask length $M = 10$.

## 7 CONCLUSION

vq-wav2vec is a self-supervised algorithm that quantizes unlabeled audio data which makes it amenable to algorithms requiring discrete data. This approach improves the state of the art on the WSJ and TIMIT benchmarks by leveraging BERT pre-training. In future work, we plan to apply

---

[3]https://github.com/drowe67/codec2

other algorithms requiring discrete inputs to audio data and to explore self-supervised pre-training algorithms which mask part of the continuous audio input. Another future work avenue is to fine-tune the pre-trained model to output transcriptions instead of feeding the pre-trained features to a custom ASR model.

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

## APPENDIX A   NUMBER OF VARIABLES VS. GROUPS

We investigate the relationship between number of variables $V$ and groups $G$. Table 6 shows that multiple groups are beneficial compared to a single group with a large number of variables. Table 7 shows that with a single group and many variables, only a small number of codewords survive.

| $V$ | 1 group | 2 groups | 4 groups | 8 groups | 16 groups | 32 groups |
|---|---|---|---|---|---|---|
| 40 | $33.44 \pm 0.24$ | $23.52 \pm 0.53$ | $18.76 \pm 0.20$ | $17.43 \pm 0.14$ | $15.97 \pm 0.21$ | $15.44 \pm 0.32$ |
| 80 | $29.14 \pm 0.70$ | $25.36 \pm 4.62$ | $17.32 \pm 0.28$ | $16.36 \pm 0.27$ | $17.55 \pm 0.27$ | $15.49 \pm 0.14$ |
| 160 | | $24.27 \pm 0.35$ | $17.55 \pm 0.03$ | $16.36 \pm 0.13$ | $15.64 \pm 0.03$ | $15.11 \pm 0.10$ |
| 320 | $27.22 \pm 0.25$ | $20.86 \pm 0.09$ | $16.49 \pm 0.07$ | $15.88 \pm 0.10$ | $15.74 \pm 0.18$ | $15.18 \pm 0.02$ |
| 640 | $26.53 \pm 2.02$ | $18.64 \pm 0.12$ | $16.60 \pm 0.22$ | $15.62 \pm 0.16$ | $15.45 \pm 0.13$ | $15.54 \pm 0.31$ |
| 1280 | $32.63 \pm 5.73$ | $18.04 \pm 0.26$ | $16.37 \pm 0.07$ | $15.85 \pm 0.05$ | $15.13 \pm 0.29$ | $15.18 \pm 0.05$ |

Table 6: PER on TIMIT dev set for vq-wav2vec models trained on Libri100. Results are based on three random seeds.

| $V$ | 1 group | 2 groups | 4 groups | 8 groups | 16 groups | 32 groups |
|---|---|---|---|---|---|---|
| 40 | 100 % (40) | 95.3 % (1.6k) | 27.4 % (2.56M) | 74.8 % (39.9M) | 99.6 % (39.9M) | 99.9 % (39.9M) |
| 80 | 92.5 % (80) | 78.5 % (6.4k) | 11.8 % (39.9M) | 91.5 % (39.9M) | 99.3 % (39.9M) | 100 % (39.9M) |
| 160 | 95 % (160) | 57.2 % (25.6k) | 35.2 % (39.9M) | 97.6 % (39.9M) | 99.8 % (39.9M) | 100 % (39.9M) |
| 320 | 33.8 % (320) | 24.6 % (102.4k) | 57.3 % (39.9M) | 98.7 % (39.9M) | 99.9 % (39.9M) | 100 % (39.9M) |
| 640 | 24.6 % (640) | 10 % (409.6k) | 60.2 % (39.9M) | 99.3 % (39.9M) | 99.9 % (39.9M) | 100 % (39.9M) |
| 1280 | 7.2 % (1.28k) | 4.9 % (1.63M) | 67.9 % (39.9M) | 99.5 % (39.9M) | 99.9 % (39.9M) | 100 % (39.9M) |

Table 7: Fraction of used codewords vs. number of theoretically possible codewords $V^G$ in brackets; 39.9M is the number of tokens in Librispeech 100h .

