# OpenReview forum: "vq-wav2vec: Self-Supervised Learning of Discrete Speech Representations"
_ICLR.cc/2020/Conference — Accept (Poster)_

### Official Review · AnonReviewer1 · 2019-10-09
**Official Blind Review #1**

**Rating:** 8

**Review:**


Overview:

This paper considers unsupervised (or self-supervised) discrete representation learning of speech using a combination of a recent vector quantized neural network discritization method and future time step prediction. Discrete representations are fine-tuned by using these as input to a BERT model; the resulting representations are then used instead of conventional speech features as the input to speech recognition models. New state-of-the-art results are achieved on two datasets.

Strengths:

The core strength of this paper is in the results that are achieved on standard speech recognition benchmarks. The results indicate that, while discritization in itself does not give improvements, coupling this with the BERT-objective results in speech features which are better in downstream speech recognition than standard features. I think the main technical novelty is in combining discritization with future time step prediction (but see the weakness below).

Weaknesses:

The main weakness of the paper is that it does not situate itself within existing literature in this area. Over the last few years, researchers in the speech community have invested significant effort in learning better speech representations, and this is not discussed. See e.g. [1]. Even more importantly, very recently there has been a number of papers investigating discrete representations of speech; see the review [2]. Some of these papers specifically use VQ-VAEs [3]. [4] actually compares VQ-VAE and the Gumbel-Softmax approach. These studies should be mentioned. This paper is different in that it incorporates future time step prediction. But context prediction has also been considered before, also for speech [5, 6, 7]. This paper can be situated as a new contribution combining these two strands of research. In the longer run it would be extremely beneficial to the community if this approach is applied to the standard benchmarks as set out in [2].

As a minor weakness, some parts of the paper is not described in enough detail and the motivation is weak or not exactly clear (see detailed comments below).

Overall assessment:

I think the results as well as the new combination of existing approaches in the paper warrants publication. But it should be amended significantly to situate itself within the existing literature. I therefore award a "weak accept".

Detailed questions and suggestions:

- Section 1: As motivation for this work, it is stated that "we aim to make well performing NLP algorithms more widely applicable". As noted above, some NLP-like ideas (such as prediction of future speech segments, stemming from text-based language modelling) have already been considered within the speech community. Rather than motivating the work in this way, it might be helpful to focus the contribution as a combination of future time step prediction and discretization (both of which have been considered in previous work, but not in combination).
- Section 4: Would it be possible to train the vq-wav2vec model jointly with BERT, i.e. as one model? I suspect it would be difficult since, for the masking objective, the discrete units are already required, but maybe there is a scheme where this could work.
- Section 2.2: Similarly to the above question, would there be a way to incorporate the BERT principles directly into an end-to-end model, e.g. by randomly masking some of the continuous input speech?
- Section 3.3: What exactly does "mode collapse" refer to in this context? Would this be using only one codebook entry, for instance?
- Section 6: It seems that in all cases to obtain improvements from discritization, BERT is required on top of the vq-wav2vec discrete symbols. Is it possible that the output acoustic model is simply better-matched to continuous rather than discrete input (direct vq-wav2vec gives discrete while BERT gives continuous)? Would it make sense to train the wav2vec acoustic model on top of the vqvae codebook entries (e) instead of directly on the symbols?

Typos, grammar and style:

- "gumbel" -> "Gumbel" (throughout; or just be consistent in capitalization)
- "which can be mitigated my workarounds" -> "which can be mitigated *by* workarounds"
- "work around" -> "workaround"

Missing references:

1. Versteegh, M., Anguera, X., Jansen, A. & Dupoux, E. (2016). The Zero Resource Speech Challenge 2015: Proposed Approaches and Results. In SLTU-2016 Procedia Computer Science, 81, (pp 67-72).
2. https://arxiv.org/abs/1904.11469
3. https://arxiv.org/abs/1905.11449
4. https://arxiv.org/abs/1904.07556
5. https://arxiv.org/abs/1904.03240
6. https://arxiv.org/abs/1807.03748 (this paper is cited)
7. https://arxiv.org/abs/1803.08976

Edit: Based on the feedback from the authors, I changed my rating from a 'weak accept' to an 'accept'.

**Experience Assessment:**

I have published in this field for several years.

**Review Assessment: Checking Correctness Of Derivations And Theory:**

I assessed the sensibility of the derivations and theory.

**Review Assessment: Checking Correctness Of Experiments:**

I assessed the sensibility of the experiments.

**Review Assessment: Thoroughness In Paper Reading:**

I read the paper at least twice and used my best judgement in assessing the paper.

---

> ### Author Response · Authors · 2019-11-15
> **Response to Reviewer #1**
>
> Thank you for the fruitful comments!
>
> We addressed your main concern and updated Section 1 of the paper to better situate it in the existing literature.
>
> >> Would it be possible to train the vq-wav2vec model jointly with BERT, i.e. as one model? [...] Similarly to the above question, would there be a way to incorporate the BERT principles directly into an end-to-end model, e.g. by randomly masking some of the continuous input speech?
>
> The focus of this paper is a quantization approach for audio. Replacing the two-step training process by an adaptation of BERT to continuous data (using a wav2vec/CPC-like objective function instead of the cross entropy) is an interesting direction for future work (and we amended the future work section accordingly). However, our current paper is a proof of concept that a pre-training scheme based on masked inputs (BERT) can improve over previous methods in the speech domain.
>
>
> >> What exactly does "mode collapse" refer to in this context?
>
> In several configurations (especially for one and two groups) considerably less codewords than theoretically possible are used. We loosely refer to mode collapse as the phenomenon when very few codewords per group are used (cf. Appendix A).
>
> We updated the paper to also refer to the appendix where we outline the number of codewords that the model uses. We observed that in the “few group regime” (G=1...4), only a few of the available centroids per group are used and refer to this phenomenon as mode collapse — for BERT training, this is actually favorable e.g. in the G=2, V=320 setting as it yields a codebook of acceptable size for NLP model training (13.5k/23k).
> Mode collapse could potentially be circumvented by strategies like embedding re-initialization used in classical k-means and this is an interesting avenue for future work.
>
>
> >> [...] BERT is required on top of the vq-wav2vec discrete symbols. Is it possible that the output acoustic model is simply better-matched to continuous rather than discrete input (direct vq-wav2vec gives discrete while BERT gives continuous)? Would it make sense to train the wav2vec acoustic model on top of the vqvae codebook entries (e) instead of directly on the symbols?
>
> We actually did what you suggest: when we train acoustic models on top of vq-wav2vec, we input the dense embedding vectors corresponding to the discrete codewords. On the other hand, we also trained an NLP sequence to sequence (Section 6.3) which takes the quantized audio codes as input and then generates the transcriptions. This gives reasonable accuracy and suggests that the discrete codes by themselves, and without the learned continuous representations, are useful. We clarified this in the updated version of the paper.
>
> We believe the reason the dense embeddings for the discrete codewords work less well is because they do not encode as much detailed context information as a representation built by wav2vec or BERT. The information in the codebook is ultimately less detailed than a context vector specific to the current input sequence.

---

### Official Review · AnonReviewer2 · 2019-10-24
**Official Blind Review #2**

**Rating:** 8

**Review:**

The paper proposes a way to pre-train quantized representations for speech. The approach proposed is a two-stage process: 1. train a quantized version of wav2vec [my understanding is that wav2vec is the same thing as CPC for Audio except for using a binary cross-entropy loss instead of InfoNCE softmax-cross entropy loss]. the authors propose to use gumbel softmax / VQ codebook for the vector quantization.
2. once you have a discrete representation, you could train BERT (as if it were a seq of language tokens). this makes a lot of sense especially given that CPC / wav2vec recovers phonemes and quantizing the phonemes will recover a language-like version of the raw audio. And running BERT across those tokens will allow you to capture the dependencies at the phoneme level.

After pre-training, the authors use the learned representations for speech recognition. They compare this to using log-mel filterbanks.

The results (WER / LER) is lower for the proposed pipeline compared to using dense wav2vec representation for n-gram and character LM.  It also makes sense that BERT helps for the k-means (vq) setting since the number of codes is large.

The authors also cleverly adopt/adapt span-BERT which is more suited to this setting.

I think this paper presents a useful contribution as far as improving speech / phoneme recognition using self-supervised learning goes, and also has useful engineering aspects in terms of combining CPC and BERT. I would like to see this paper accepted.

**Experience Assessment:**

I have published one or two papers in this area.

**Review Assessment: Checking Correctness Of Derivations And Theory:**

I carefully checked the derivations and theory.

**Review Assessment: Checking Correctness Of Experiments:**

I assessed the sensibility of the experiments.

**Review Assessment: Thoroughness In Paper Reading:**

I read the paper at least twice and used my best judgement in assessing the paper.

---

> ### Author Response · Authors · 2019-11-15
> **Response to Reviewer #2**
>
> Thank you for your comments!

---

### Official Review · AnonReviewer3 · 2019-10-30
**Official Blind Review #3**

**Rating:** 6

**Review:**

Though rather dense in its exposition, this paper is an interesting contribution to the area of self-supervised learning  based on discrete representations. What would make it stronger imo is to address the issue of how much is gained from a discrete vs. continuous representation. The authors take it as a given that discrete is good because it allows us to leverage work in NLP. That makes sense -- but at what cost?

"Table 4 shows that our first results are promising, even though they are not as good as the state of the art." The state of the art on LibriSpeech is not Mohamed at al. 2019. See e.g. Irie et al. Interspeech 2019 for better result

The Conclusion is very sparse. "In future work, we are planning to apply other algorithms requiring discrete inputs to audio data": can  you elaborate?

**Experience Assessment:**

I have read many papers in this area.

**Review Assessment: Checking Correctness Of Derivations And Theory:**

I assessed the sensibility of the derivations and theory.

**Review Assessment: Checking Correctness Of Experiments:**

I assessed the sensibility of the experiments.

**Review Assessment: Thoroughness In Paper Reading:**

I made a quick assessment of this paper.

---

> ### Author Response · Authors · 2019-11-15
> **Response to Reviewer #3**
>
> Thank you for your fruitful comments.
>
> >> What would make it stronger imo is to address the issue of how much is gained from a discrete vs. continuous representation.
> Discrete representations by themselves are not better than continuous ones (cf. Table 1, wav2vec vs. vq-wav2vec). However, discretization enables the application of existing algorithms from the NLP literature which were designed for discrete inputs. We show that the BERT model can be directly applied to discretized speech. BERT can better model context than (vq-)wav2vec.
>
> >> The authors take it as a given that discrete is good because it allows us to leverage work in NLP. That makes sense -- but at what cost?
> Chaining vq-wav2vec and BERT requires more computational effort than just wav2vec, however, it does improve accuracy as our results show (cf. Table 1). Running BERT requires roughly as much computational overhead as just vq-wav2vec.
>
> >> The state of the art on LibriSpeech is not Mohamed at al. 2019. See e.g. Irie et al. Interspeech 2019 for better result.
> Thanks for pointing this out, we fixed this in the updated version of the paper we just posted.
>
> >> The Conclusion is very sparse.
> We broadened conclusion and delineated additional future work.

---

### Official Review · AnonReviewer4 · 2019-11-03
**Official Blind Review #4**

**Rating:** 8

**Review:**

This paper presents a method for unsupervised representation learning of speech. The idea is to first learn discrete representation (vector quantization is done by Gumbel softmax or k-means) from audio samples with contrastive prediction coding type objective, and then perform BERT-style pre-training (borrowed from NLP). The BERT features are used as inputs to ASR systems, rather than the usual log-mel features. The idea, which combines those of previous work (wav2vec and BERT) synergetically, is intuitive and clearly presented, significant improvements over log-mel and wav2vec were achieved on ASR benchmarks WSJ and TIMIT. Based on these merits, I suggest this paper to be accepted.

On the other hand, I would suggest directions for investigation and improvements as follows.

1. While I understand that vector quantization makes the use of NLP-style BERT-training possible (as the inputs to NLP models are discrete tokens),  there are potential disadvantages as well. One observation from the submission is that the token set may need to very large (from tens of thousands to millions) for the system to work well, making the BERT training computationally expensive (I noticed that the BERT model is trained on 128 GPUs). Also, without BERT pre-training, using directly the discrete tokens seems to consistently give worse performance for ASR. I think some more motivations or explorations (what kind of information did BERT learn) are needed to understand why that is the case.

2. Besides the computational expensive-ness of the three-step approach (vector quantization, BERT, acoustic model training), the combined model complexity is large because these steps do not share neural network architecture. A more economical approach is to use BERT-trained model as initialization for acoustic model training, which is the classical way how RBMs pre-training were used in ASR.

3. One concern I have with discrete representation is how robust they are wrt different dataset. The ASR datasets used in this work are relatively clean (but there does exists domain difference between them). It remains to see how the method performs with more acoustically-challenging speech data, and how universally useful the learned features are (as is the case for BERT in NLP).

4. Another curious question is whether the features would still provide as much improvement when a stronger ASR system than AutoSeg (e.g., Lattice-free MMI) is used.

Overall, while I think the computational cost of the proposed method is high, rendering it less practical at this point, I believe the approach has potential and the result obtained so far is already significant.

**Experience Assessment:**

I have read many papers in this area.

**Review Assessment: Checking Correctness Of Derivations And Theory:**

N/A

**Review Assessment: Checking Correctness Of Experiments:**

I assessed the sensibility of the experiments.

**Review Assessment: Thoroughness In Paper Reading:**

I read the paper at least twice and used my best judgement in assessing the paper.

---

> ### Author Response · Authors · 2019-11-15
> **Response to Reviewer #4**
>
> Thank you for your fruitful comments.
>
> >> 1. [...]  One observation from the submission is that the token set may need to be very large (from tens of thousands to millions) for the system to work well, making the BERT training computationally expensive [...] I think some more motivation or exploration (what kind of information did BERT learn) is needed to understand why that is the case.
>
> Our BERT vocabulary sizes (13.5k for the gumbel version and 23k for the k-means version) compare favorably to the setups commonly used in NLP where vocabularies are double or triple of our sizes.
>
> We agree that it would be interesting to perform an in-depth analysis on the embeddings learned by BERT and we will investigate this in future work. Here we focus on a new quantization method evaluated via downstream performance in phone and speech recognition settings by employing models that worked well (and were extensively tuned) in NLP contexts.
>
>
> >> 2. A more economical approach is to use BERT-trained model as initialization for acoustic model training, which is the classical way how RBMs pre-training were used in ASR.
>
> Yes, this is an interesting avenue for future work! We did not follow this direction due to two motivations: first, our aim is to contribute a new quantization scheme for audio data that is trained to predict the context in a self-supervised way. Second, we wanted to show that good performance can be achieved with discretized audio on actual speech tasks.
>
>
> >> 3. One concern I have with discrete representation is how robust they are wrt different dataset.
> We agree that an ablation study on robustness of the embeddings across different datasets would be very interesting.
>
> Here we are mostly focusing on relatively clean data (WSJ, TIMIT, Librispeech) following the original wav2vec paper but we would be interested in exploring robustness in the future. However we note that representations transfer at least well across datasets from the “clean speech” domain: vq-wav2vec and BERT is only trained on Librispeech and never tuned on TIMIT/WSJ.
>
> >> 4. Another curious question is whether the features would still provide as much improvement when a stronger ASR system than AutoSeg (e.g., Lattice-free MMI) is used.
>
> The original wav2vec paper (Schneider et al., 2019) reports better results than LF-MMI on the WSJ benchmark, however, the two setups are not strictly comparable. In some sense, the LF-MMI result has an edge because it is based on a phoneme-based ASR system which is typically stronger than the character-based ASR system used with wav2vec. We agree that evaluation on stronger baselines is an important future direction though.

---

### Author Response · Authors · 2019-11-15
**Paper updates**

We just updated the paper to incorporate the reviewer comments. Specifically, the updated version includes:

* Improved discussion of related work and better situation of our contribution in the existing literature
* Extended conclusion & future work
* Improved results for sequence to sequence learning (Table 4) + more results from the literature, e.g., Irie et al. ‘19
* For the vq-wav2vec ASR experiments in Section 6.1, we clarified that we input the dense representations associated with the discrete units.

Big thank you to the reviewers for their comments!

---

### Decision · Program_Chairs · 2019-12-19

**Decision:**

Accept (Poster)

**Comment:**

This paper proposes a new self-supervised pre-trained speech model that improves speech recognition performance.
 The idea combines an earlier pre-training approach (wav2vec) with discretization followed by BERT-style masked reconstruction.  The result is a fairly complex approach, with not too much novelty but with a good amount of engineering and analysis, and ultimately very good performance.  The reviewers agree that the work deserves publication at ICLR, and the authors have addressed some of the reviewer concerns in their revision.  The complexity of the approach may mean that it is not immediately widely adopted by others, but it is a good proof of concept and may well inspire other related work.  I believe the ICLR community will find this work interesting.